# Effects of Various Marine Toxins on the Mouse Intestine Organoid Model

**DOI:** 10.3390/toxins14120829

**Published:** 2022-11-24

**Authors:** Nengzhuang Wang, Minghua Liu, Fengrui Bi, Long Ma, Lina Qin, Yao Wang, Kai Gu, Xuemei Ge, Hongli Yan

**Affiliations:** 1Reproductive Medicine Center, The First Affiliated Hospital of China Naval Military Medical University, Shanghai 200433, China; 2Shanghai Outdo Biotech Co., Ltd., Shanghai 200433, China

**Keywords:** marine toxins, okadaic acid, conotoxin, mouse intestine organoid

## Abstract

Because of their trace existence, exquisite structure and unique role, highly toxic marine biotoxins have always led to the development of natural product identification, structure and function research, chemistry and biosynthesis, and there are still many deficiencies in the injury and protection of highly toxic organisms, toxin biosynthesis, rapid detection, poisoning and diagnosis and treatment. In this study, a mouse intestine organoid (MIO) model was constructed to explore the effects of the marine toxins okadaic acid (OA) and conotoxin (CgTx) on MIO. The results showed that the cell mortality caused by the two toxins at middle and high concentrations was significantly higher than the cell mortality of the control group, the ATPase activity in each group exposed to OA was significantly lower than the ATPase activity of the control group, all the CgTx groups were significantly higher than that of the control group, and the number of apoptotic cells was not significantly higher than the number of apoptotic cells of the control group. Through RNA-Seq differential genes, Gene Ontology (GO) and pathway analysis, and Gene Set Enrichment Analysis (GSEA) experimental results, it was demonstrated that OA reduced cell metabolism and energy production by affecting cell transcription in MIO. Ultimately, cell death resulted. In contrast, CgTx upregulated the intracellular hormone metabolism pathway by affecting the nuclear receptor pathway of MIO, which resulted in cell death and the generation of energy in large amounts.

## 1. Introduction

Due to their trace existence, exquisite structure and unique role, highly toxic marine biotoxins have always led to the development of natural product identification, structure and function research, chemistry and biosynthesis [1,2,3]. Injury and poisoning can be caused by direct exposure or attack by poisonous organisms, ingestion of poisoned food or inhalation of toxic fumes, characterized by instant effects and difficulty in detection and treatment [3,4,5]. It has now become a common problem in many fields, such as ecological monitoring, food detection and pharmacological and toxicological research [6,7]. Currently, there are still many deficiencies in the injury and protection of highly toxic organisms, toxin biosynthesis, rapid detection, poisoning and diagnosis and treatment [8,9]. Diarrheal shellfish poisoning is a foodborne disease caused by the ingestion of seafood contaminated by toxins [1]. Okadaic acid (OA) and its derivative dinophysistoxins (DTXs) are the most common and detrimental components of diarrhetic shellfish poison (DSP) [10]. OA and its derivatives have now been found to trigger symptoms such as diarrhea, nausea and vomiting by inhibiting protein phosphorylation [11]. In addition, these toxins can produce various chronic toxic effects. In some studies, OA toxins have been shown to exhibit cytotoxicity, genotoxicity, neurotoxicity, and immunotoxicity [12]. As a highly active biological polypeptide toxin secreted by marine carnivorous CgTx, CgTx can be used to kill prey [13]. Approximately 1 million different varieties of CgTx are estimated to be distributed around the world [14]. According to the conserved signal region, these CgTx can be divided into 27 superfamilies, such as A, B2, C, D, O, M, and T [15]. For each of them, the target ion channel and receptor differ [16]. Therefore, they have become lead compounds with potential medicinal value and molecular markers for neuropharmacology [17]. The reported conotoxins targeting nicotinic acetylcholine receptors are derived from 10 superfamilies, namely, A, B3, C, D, J, L, S, O1, M, and T [18].

Organoids refer to organ-specific cell cultures that develop from pluripotent stem cells or organ progenitor cells [19,20]. These cells combine organically in vitro in a similar way to that in vivo to form three-dimensional (3D) cultures with certain structures and functions [20]. They are characterized by spatial organization and cellular functions similar to those of the natural tissues simulated by them [21]. In recent years, research has been conducted to build many human and mammalian organoid models, including organoids representing the intestine, prostate, ovary, bladder, liver, and brain [20,22,23,24,25,26,27,28,29,30,31]. For example, it has been demonstrated in cell experiments in vitro that organoid models are fit for drug efficacy and toxicity assessment [31]. Furthermore, organoid models have been proven to be capable of simulating functional units in vivo by building an “organoid chip” for drug screening and disease-model construction [31,32]. Therefore, organoids play an essential role in regenerative medicine, drug research, and gene therapy, linking in vivo and in vitro experiments [20,31].

In this study, the MIO model was successfully constructed to explore the effects of different concentrations of OA and CgTx (μ-conotoxin CnIIIC), as the representative toxins of DSP and biological polypeptide toxin from marine toxins, on cell mortality, ATPase activity, and apoptosis in MIO. Then, based on the above experimental results, high-throughput RNA-Seq was used to detect the differentially expressed genes, GO terms and pathways and Gene Set Enrichment Analysis (GSEA) in MIOs exposed to OA and CgTx. Notably, it is a promising and effective solution to study the effects of the marine toxins OA and CgTx on the digestive system through the MIO model. According to our research results, MIO is applicable as an effective biological experimental model for marine toxins OA and CgTx.

## 2. Results

### 2.1. Construction of MIO Model

The MIO model shown in Figure 1 was constructed using the medium of SHANGHAI OUTDO BIOTECH and Matrigel and then identified by immunofluorescence experiments to illustrate the epithelial cells, intestinal stem cells, Paneth cells, endocrine cells, and tuft cells (Figure 2). The results suggest the successful construction of the MIO model.

### 2.2. Detection of MIO Activity after Exposure to Marine Toxins

The MIO model was adopted to detect the effect of marine toxins on the activity of MIO (Figure 3). The data on cell mortality were gathered after exposure to marine toxins in the MIO. After the introduction of OA for 2 days, the cell mortality of the 5 μM and 10 μM groups was found to be significantly different from the mortality of the control group (*p* < 0.01, *p* < 0.01). For the one-way analysis of variance (ANOVA), *p* < 0.01, and IC_50_ was 3.611 μM. After the introduction of CgTx for 2 days, there was a significant difference found in cell mortality between the 4-μM and 20-μM groups and the control group (*p* < 0.05, *p* < 0.0001). One-way ANOVA, *p* < 0.0001, IC_50_ was 7.474 μM (Figure 4).

### 2.3. Detection of ATPase Activity in MIO after Exposure to Marine Toxins

The MIO model was used to detect the effect of marine toxins on the ATPase activity of MIO (Figure 5). After the introduction of OA for 2 days, the ATPase activity in each treatment group was found to be significantly lower than the ATPase activity in the control group (*p* < 0.0001), one-way ANOVA, *p* < 0.0001. In addition, 2 days after the addition of CgTx, the ATPase activity in each treatment group was found to be significantly higher than the ATPase in the control group (*p* < 0.0001), one-way ANOVA, *p* < 0.0001 (Figure 5).

### 2.4. Detection of Apoptosis in MIOs after Exposure to Marine Toxins

The MIO model was applied to detect the effect of marine toxins on MIO apoptosis (Figure 6). After the introduction of OA for 2 days, there was no significant difference found in apoptosis between each treatment group and the control group (one-way ANOVA, *p* > 0.05). In addition, there was no significant difference found in apoptosis between each treatment group and the control group after the introduction of CgTx for 2 days (one-way ANOVA, *p* > 0.05) (Figure 6).

### 2.5. High-Throughput mRNA-Seq of MIOs after Exposure to Marine Toxins

#### 2.5.1. Differential Gene Analysis

The original result of Illumina sequencing is the original image file. Then, after base recognition and error filtering, the original sequencing fragment used for analysis is obtained, referred to as reads. Data Q-value box chart (box plot): A box chart is a statistical chart intended to show the distribution of data. The Q-value box chart is the quality quantile chart. The lowest edge of the yellow rectangle is the Q-value 1/4 quantile, while the upper and lower black lines account for 3/4 of the corresponding quality values, respectively. The blue lines represent the average value of the mass. Different background colours indicate the quality of this part, the green background represents the high quality value part, the orange background indicates the reasonable quality value part, and the red background represents the low quality value part, as shown in Appendix A. Base distribution map: According to the principle of base complementary pairing, GC and AT base pairs conform to a uniform distribution, and the base distribution balance is evaluated against the base distribution map. Due to the random primers used to construct the library, the first few bases will show a preference in nucleotide composition to some extent, thus resulting in normal fluctuations. Then, it tends to stabilize, as shown in Appendix A. The original result of Illumina sequencing is the original image file. Then, after base recognition and error filtering, the original sequencing fragment used for analysis is obtained, which is referred to as reads. Stored in fastq format, the result includes the base composition information of the sequence and its corresponding information about sequence quality. Double-ended sequencing (PE) was divided into two read files: _ R1MagneR2 (Appendix A). The preprocessed sequencing sequence was used for genomic mapping with the assistance of HISAT2 software. Based on global and local search methods, HISAT2 can be applied to carry out mapping efficiently and perform comparisons with the spliced reads in RNA-Seq sequencing data effectively with the default parameter set. The R-packet edgeR was employed to calculate the expression difference of the expression count matrix according to the grouping information (Table 1 and Table 2). In addition, the different mRNAs corresponding to fragments per kilobase of transcript per million mapped reads (FPKM) values were selected for clustering heatmap visualization (Figure 7). The results are listed in Table 1 and Table 2.

#### 2.5.2. GO and Pathway Analysis

The R package was applied to analyse the differential miRNA target genes by Gene Ontology (GO) and pathway analysis using Fisher’s exact test (Appendix A, Table 3, Table 4, Table 5 and Table 6). The results are listed in Table 3, Table 4, Table 5 and Table 6.

#### 2.5.3. GSEA

According to the GSEA of the expression dataset, the gene data involved 7970 genes. In the selected pathway database, this gene set involved 2232 pieces of pathway functional information. In line with the defined selection criteria (the number of genes in the pathway subset was above 5 and below 10,000), only 1948 pathway base factor sets were used for analysis. Among those genes positively related to OA and the 1948 pathway studied, there were 753 genes in the pathway showing a positive correlation with OA. In addition, 75% of the correct rate supported that no genes in the pathway were positively correlated and that 42 genes in the pathway were positively correlated with *p* < 0.01. Among the genes negatively related to OA, 1195 genes in the pathway showed a negative correlation, and 75% of the correct rate supported the negative correlation of gene expression in 8 of them in the pathway, with the negative correlation *p* value falling below 0.01 for the gene expression in the pathway (Appendix A). According to the GSEA of the expression dataset, the gene data involved 2985 genes. In the selected pathway database, this gene set involved 2232 pieces of pathway functional information. In line with the defined selection criteria (the number of genes in the pathway subset was above 5 and below 10,000), only 1151 pathway base factor sets were used for analysis. Among the genes positively related to CgTx and the 1151 pathway studied, 560 genes in the pathway were positively correlated with CgTx, 75% of the correct rate supported the positive correlation of gene expression in none of them in the pathway, and the *p* value of positive correlation in 65 pathways fell below 0.01. Among the genes negatively related to CgTx, 591 genes in the pathway showed a negative correlation, 75% of the correct rate supported the negative correlation of gene expression in 1 pathway, and the negative correlation *p* value fell below 0.01 for 45 pathway genes (Appendix A).

## 3. Discussion

The MIO model was constructed using MIO medium provided by SHANGHAI OUTDO BIOTECH. In a previous preexperiment, a comparison was performed between MIO culture medium and Stemcell^®^ medium, revealing that MIO medium sourced from SHANGHAI OUTDO BIOTECH Company is advantageous in organoid subculture [33]. In the detection of cell mortality in MIO exposed to marine toxins, the level of cell mortality was found to increase significantly at middle and high concentrations of the two toxins relative to the control group. However, the ATPase activity in all groups exposed to OA was found to be significantly lower than the ATPase activity in the control group, while ATPase activity in all CgTx groups was found to be significantly higher than the ATPase activity in the control group [17,18,34]. This observation is suspected to result from the fact that CgTx affects hormone metabolism in MIO [34]. The results of the terminal deoxynucleotidyl transferase dUTP nick end labelling (TUNEL) experiment suggested no significant increase in the number of apoptotic cells in each treatment group in comparison with the control group.

According to our top 20 analyses of RNA-Seq differential genes, OA affected MIO mainly by altering the expression of nervous system-related genes, the function of RNAase II transcription binding DNA, and the gene expression of the Wnt pathway, encoding transcriptional initiation-related proteins, and influencing haem activity. CgTx treatment of MIO exerted effects mainly by altering the transcriptional function, the function of ion channel, the function of DNA replication changed, the gene expression of Wnt pathway, and ligand binding. As revealed by GO and pathway analysis, OA affected mainly the mitochondrial inner membrane, organelle inner membrane, mitochondrial protein-containing complex and other relevant pathways in MIO. By comparison, CgTx affected mainly chemical carcinogenesis–reactive oxygen species (ROS), prion disease, peroxisome and other related pathways in MIO, consistent with the results obtained from our ATPase activity experiment. According to the GSEA results, OA significantly downregulated the synthesis of proteins related to transcription initiation in MIO, while CgTx significantly downregulated the nuclear receptor pathway and upregulated the hormone metabolism pathway MIO. OA can reduce cell metabolism and energy production by affecting cell transcription in MIO, thus resulting in cell death, while CgTx is capable of upregulating the intracellular hormone metabolism pathway by affecting the nuclear receptor pathway of MIO, thus resulting in cell death and generating energy in large amounts [17,18,34,35,36,37,38,39].

## 4. Conclusions

In summary, an MIO model was constructed to explore the effects of the marine toxins OA and CgTx on MIO. According to the research results, the cell mortality caused by the two toxins at middle and high concentrations was significantly higher compared to the control group, while the activity of ATPase in each group exposed to OA was significantly lower than the activity of ATPase in the control group, while all the CgTx groups were significantly higher than the ATPase activity of the control group, while the number of apoptotic cells was not significantly increased compared with the control group. Through RNA-Seq differential genes, GO and pathway analysis, and GSEA experimental results, OA was discovered to reduce cell metabolism and energy production by affecting cell transcription in MIO, which led to cell death, and that CgTx upregulated the intracellular hormone metabolism pathway by affecting the nuclear receptor pathway of MIO, which led to cell death and the generation of energy in large amounts.

## 5. Materials and Methods

### 5.1. Primary Culture of MIO

After the mice were killed, the proximal gastric end 20 cm of the small intestine of the mice was removed and placed in a 10-cm Petri dish. After repeated washing with phosphate-buffered saline (PBS) washing solution, the gastric end 20 cm was cut longitudinally in a 10-cm Petri dish and washed repeatedly with PBS. After transfer to a 50-mL centrifuge tube and digestion at 4 °C for 30 min with 0.02 PBS, the small intestinal epithelium was scraped with cover slides, and fresh MEDTA was added for even blowing. After sieving (100 μm), the filtrate was screened (40 μm), and the small intestinal recess failing to pass through the screen was resuscitated with PBS. Then, the cells were washed and centrifuged in medium without cytokines (1000 rpm for 3 min) for cell precipitation. Next, 40 μL of suspension cells was added to the cell precipitate (approximately 30–400,000 cells), 260 μL of Matrigel was added to the ice, and 50 μL of cell suspension was added to the 24-well plate. After the culture plate was transferred to the cell incubator at 37 °C for 15 min, 600 μL of medium (SHANGHAI OUTDO BIOTECH)/well after Matrigel solidification was added. The fluid was changed every 2–3 days.

### 5.2. Subculture of MIO

On the day of passage, the culture plate was removed from the cell incubator at 37 °C, repeatedly incubated with a liquid transfer gun, dispersed, collected, and centrifuged (1000 rpm for 3 min), and the excess supernatant was removed to obtain the cell precipitate. The cell precipitates were obtained by washing and centrifugation in medium without cytokines (1000 rpm for 3 min). Forty microlitres of suspension cells were added to the cell precipitate (approximately 30–400,000 cells), 260 μL Matrigel was added to the ice, and 50 μL cell suspension was added to the 24-well plate. The culture plate was transferred to a cell incubator at 37 °C for 15 min, and 600 μL medium/well was added after Matrigel solidification. The fluid was changed every 2–3 days.

### 5.3. Identification of MIO

After the collection of organoids, the supernatant was centrifuged to remove the supernatant, and cell recovery reagent was added for gelation. The whole process was carried out on ice for 1 h. Then, the supernatant was centrifuged, fixed with 4% paraformaldehyde overnight at 4 °C, removed the following morning, washed three times with PBS, gently removed, and centrifuged to remove the supernatant. Next, 3% agarose was applied as a scaffold at the bottom of the centrifuge tube to mix agarose and similar organs evenly for addition into the agarose scaffold. Then, the samples were frozen for 30 min in the refrigerator, paraffin-embedded, sliced and spread. Subsequently, the slides were placed in an oven, with the temperature set to 63 °C, and the wax was dried for one hour. PBS buffer (formula: 80 g NaCl, 2 g KCl, 15.35 g Na_2_HPO_4_, 2 g KH_2_PO_4_, fixed volume to 1000 mL pH 7.2 with pure water): 10 × PBS buffer was diluted to 1 × PBS buffer, and then 0.05% Tween reagent was added into 1 × PBS buffer. Citric acid repair solution: A sodium citrate solution of 82 mL 0.1 mol/L + citric acid solution of 18 mL 0.1 mol/L + pure water of 900 mL was placed in a pressure cooker (pH = 5.96). Ethylenediaminetetraacetic acid (EDTA) repair solution: pH = 9.0. The 50 × EDTA antigen repair solution (Maixin) was diluted 50 times with pure water. After being baked, the film was removed from the oven and placed into the automatic dyeing machine for dewaxing. The dewaxing process is detailed as follows: xylene (two cylinders), each cylinder 15 min (according to the instrument setting time); anhydrous ethanol two cylinders, each cylinder 7 min (according to the instrument setting time); 90% alcohol 1 cylinder, 5 min (according to the instrument setting time); 80% alcohol 1 cylinder, 5 min (according to the instrument setting time). 70% alcohol 1 cylinder, 5 min (according to the instrument setting time). The film was removed from the dyeing machine and rinsed with pure water 3 times for no less than 1 min each time. In the process of flushing, citric acid repair solution or EDTA repair solution was added to the induction cooker for heating. Citric acid high-pressure repair: After the citric acid repair solution was boiled, the film was placed into a pressure cooker, and the high-pressure pot was covered. When the time was up, heating was stopped, the cover of the high-pressure pot was opened, and the pot was allowed to cool down naturally for more than 30 min. EDTA hot repair: The solution was put into the film after heating and boiling for 20 min. By using a commercial ready-to-use blocker, the blocker was dropped onto the tablet for 10–15 min. Then, the tablet was removed and rinsed with PBS buffer for 1 min at a time. An antibody was removed from the refrigerator and placed in a centrifuge for no less than 30 s. Then, the antibody was removed and diluted with an antibody diluent according to the requirements of the customer or the antibody manual. The antibody was ground for dilution according to the customer’s or antibody manual; if the customer did not require it, the dilution was 1:200/1:500. (1) Epithelial cells and positive E-cadherin (Abcam, Cambridge, UK) staining, (2) stem cells (Intestinal stem cells), positive for proliferating cell nuclear antigen (PCNA) (Abcam, Cambridge, UK) staining, (3) Paneth cells, positive for lysozyme (Abcam), (4) endocrine cells, positive for Chromogranin A (Abcam) staining, (5) clusters of cells (Tuft cells), positive for double cortical protein-like kinase (Doublecortin-like kinase, DCLK)-1 (Abcam), incubated at room temperature for 30 min or refrigerated overnight at 4 °C, and the tablets were rinsed with PBS buffer 3 times, one minute at a time. For slides overnight at 4 °C, the slides were reheated at room temperature for more than 30 min and then rinsed with PBS washing solution before the experiment, and the second antibody ready-to-use working solution was added and incubated at room temperature for 30 min. When the time was up, the slide was rinsed with PBS 3 times for not less than 1 min each time. After 3,3′-diaminobenzidine (DAB) colour development, the DAB kit was removed from the refrigerator and prepared according to 1 mL DAB diluent + 1 drop of DAB chromophore. Diluted DAB was added to the film to observe the colour intensity, with the longest colour intensity reaching 5 min. Then, the film was rinsed with tap water for 5 min. After the addition of haematoxylin (SIGMA, Burlington, MA, USA) to the tablet for 1 min, it was immersed in 0.25% hydrochloric acid in alcohol (400 mL 70% alcohol + 1 mL concentrated hydrochloric acid) for no less than 2 s, rinsed with tap water for more than 2 min, and dried at room temperature to seal the tablet [19,33,40,41].

### 5.4. Preparation and Exposure of Marine Toxins

The concentration gradient of the MIO exposed to two kinds of toxins was designed as follows. The low, middle and high concentrations of OA (Puhuashi Technology Development Co., Ltd., Beijing, China, A-OA-C100, Purity ≥ 95% by HPLC) were 0, 1, 5 and 10 μM, respectively. The low, middle and high concentrations of CgTx (μ-conotoxin CnIIIC) (APeptide Co., Ltd., Sunnyvale, CA, USA, P201028-K6, Purity ≥ 95.13% by HPLC) were 0.4, 4 and 20 μM, respectively. MIO was inoculated into 96-well plates and then exposed according to the above concentration gradient after 14 days of subculture. The exposure time was set to 48 h, and the control group was set up at the same time [1,7,10,12,13,17,34,42,43,44,45].

### 5.5. MIO Activity Experiment

The cells were cultured in 96-well plates, and the exposure concentration and time were set to the same as above. After exposure, the living and dead cells were stained with a Calcein AM/PI Double Stain Kit (MKBio, Yamaguchi, Japan) and a dead cell double staining kit [46]. Then, the cells were collected by centrifugation, 1000 Assay Buffer for 3 min, the supernatant, thoroughly washed the cells with 1 × Assay Buffer for 2–3 times, prepared with 1 × Assay Buffer cell suspension with a density of 1 × 10~5–1 × 10~6 cells/mL, 100 μL staining working solution added to 200 μL cell suspension, mixed and incubated at 37 °C for 15 min. 490 ± 10 nm excitation filter under fluorescence microscope, detection of living cells (green) and dead cells (red), and calculation of cell mortality and IC 50, such as Figure 1 and Figure 2.

### 5.6. ATPase Activity Assay

The CellTiter-Glo^®^ (Promega, Madison, WI, USA) kit was used to detect the activity of MIO exposed to culture for 48 h [19]. The 96-well plate organoid medium was removed, and 100 μL of fresh nontoxic F12 was added. A control hole containing 100 μL of acellular F12 per hole was prepared to obtain the background luminescence value. With the addition of CellTiter-Glo^®^ reagent with a volume equal to the volume of cell culture medium present in each well, an all-black 96-well plate was used. The mixed content was shaken for 2 min to induce cell lysis. The plate hatch was left at room temperature for 10 min to stabilize the luminous signal. Then, the luminescence was recorded by an enzyme labelling instrument, and the relative activity of the cells was calculated through an ATP standard curve.

### 5.7. TUNEL

Cells (2 million cells at most) were collected and washed once with PBS or Hanks’ balanced salt solution (HBSS). Then, the cells were fixed with immune staining fixation solution (P0098) produced by Biyuntian or 4% paraformaldehyde for 30 min. To prevent the cells from agglomerating to form a mass, they should be fixed while shaken gently on the lateral shaking table or horizontal shaking table. After being washed once with PBS or HBSS, the cells were incubated for 5 min with the immunostaining strong permeable solution (P0097) produced by Biyuntian or PBS resuspension cells containing 0.3% Triton X-100 at room temperature. Thorough mixing was required for the preparation of TUNEL (Beyotime, Jiangsu, China) detection solution for each sample, 5 μL of TdT enzyme, and 45 μL of fluorescence labelling solution. Wash twice with PBS or HBSS. Fifty microlitres of TUNEL detection solution was added and incubated for 60 min at 37 °C. Wash twice with PBS or HBSS. Suspension with 250–500 μL PBS or HBSS. At this point, it was detected by flow cytometry or smeared under a fluorescence microscope (LEICA DMI3000 B, ×50). The excitation wavelength range was 450–500 nm, and the emission wavelength range was 515–565 nm (green) [33]. Apoptotic cells were counted with the image scope×64 software.

### 5.8. High-Throughput mRNA-seq

#### 5.8.1. Background Introduction

MIO was inoculated into 96-well plates and exposed to 5 μM OA and 4 μM CgTx after 14 days of subculture. The exposure time was set to 48 h. The subject of mRNA sequencing comprises all the gene messenger RNAs that can be transcribed by a particular cell in a certain functional state. Transcriptome studies play an essential role in the study of gene function and structure. Through new-generation sequencing, it is possible to determine the expression abundance of almost all mRNAs in a specific tissue or organ of a species in an accurate and fast way. Currently, high-throughput second-generation sequencing has been widely practised in many fields, such as basic research and drug development [40,47,48,49,50].

#### 5.8.2. Database Building Process

The mRNA sequencing experiment starts by extracting RNA from samples. Due to the degradation of RNA, the whole experimental process must be conducted rigorously to ensure the stability and consistency of the data. Appendix A shows more information on the overall database construction process of second-generation sequencing.

##### Total RNA Quality Inspection

An RNeasy Mini Kit (Cat#74106, Qiagen) was used for total RNA extraction from the samples according to the instructions provided by the manufacturer. The total RNA was inspected by using a Nano Drop ND-2000 spectrophotometer and Agilent Bioanalyzer 4200 (Agilent Technologies, Santa Clara, CA, USA), while the RNA passing the quality inspection was used to construct the library.

##### Library Construction and Quality Inspection

For the specific eukaryotic species, magnetic beads were used to remove rRNA. Then, the RNA without rRNA was tested through fragmentation, double-stranded cDNA synthesis, second-chain degradation, end repair, the addition of An at the 3′ end, connection and amplification. In the constructed library, a Qubit ^®^ 2.0 fluorometer was used to determine the concentration, and an Agilent 2100 was used to detect the size.

##### Computer Sequencing

The quality inspection qualified library was ready for Illumina sequencing, and the sequencing strategy was PE150. The basic principle of sequencing is sequencing by synthesis (SBS): the flowcell carrying a cluster is put on the computer, with four kinds of fluorescently labelled dNTPs, DNA polymerase and splice primers added to the flowcell for amplification. When each sequencing cluster was used to extend the complementary chain, each fluorescently labelled dNTP released the corresponding fluorescence. The sequencer captures the fluorescence signal and converts the optical signal into the sequencing peak through computer software (Appendix A). As a result, sequence information was obtained for the fragments to be tested (Appendix A).

#### 5.8.3. RNA Sequencing and Differentially Expressed Genes Analysis

The libraries were sequenced on an software trim_galore (version 0.6.4) platform and 150 bp paired-end reads were generated. About 53.15 millions raw reads for each sample were generated (Appendix A). Raw reads of fastq format were firstly processed using trim_galore (version 0.6.4) to remove adaptor sequences and low quality reads and Bowtie2 software (version 2.4.2) was used to align to the rRNA database to remove rRNA reads. Then about 53.04 millions clean reads for each sample were retained for subsequent analyses (Appendix A). The clean reads were mapped to the reference genome Mouse: GRCm39.104 using hisat2 (version 2.2.0) software (Appendix A). FPKM of each gene was calculated and the read counts of each gene were obtained by htseq-count (version 0.13.5).

Differential expression analysis was performed using the edgeR. Q value < 0.05 and foldchange > 2 or foldchange < 0.5 was set as the threshold for significantly differential expression gene (DEGs). Based on the hypergeometric distribution, Gene Ontology (GO) and Kyoto Encyclopedia of Genes and Genomes (KEGG) pathway enrichment analysis of DEGs were performed to screen the significant enriched term using R (v 3.2.0), respectively. Gene Set Enrichment Analysis (GSEA) was performed using GSEA software. The analysis was used a predefined gene set, and the genes were ranked according to the degree of differential expression in the two types of samples. Then it is tested whether the predefined gene set was enriched at the top or bottom of the ranking list (Appendix A).

#### 5.8.4. Quantitative Results

##### Quantification and Visualization of Sample RNA

The software htseq-count was used to quantify the RNA of the data, and the expression quantity (count) was calculated for each molecule in each sample. Then, the FPKM (fragments per kilobase of exon model per million mapped reads) value of the corresponding molecule in each sample was calculated to assess the common method of gene expression, which offsets the deviation caused by the variation in gene length when the expression quantity is represented by read number. In the meantime, the difference in the amount of sequencing data will also affect the number of reads. The formula is as follows: the number of fragments compared to gene exons; ExonLength: the total length of gene exons; MappedReads: the total number of reads compared to the reference genome.

##### Quantitative Result Display

Statistics of Gene Expression Distribution

The box map of all gene expression values (FPKM) of each sample was built to show the distribution of gene expression levels in different samples, where the abscissa is the sample name and the ordinate is log_2_ (FPKM + 1). The box chart is composed of quartiles, the maximum, the upper quartile, the median, the lower quartile and the minimum from top to bottom. The two ends of the box diagram rectangle box correspond to the upper and lower quartiles (Q3 and Q1), respectively. The internal median corresponds to the median line, the upper line is Q1 + 1.5IQR, and the underline is Q3 − 1.5IQR, where IQR (InterQuantileRange) represents the middle quartile range, and IQR = Q3-Q1, as shown in Appendix A.

2.Correlation

The Pearson correlation coefficient of each sample was calculated and visualized to show the repeatability between groups, such as the log_2_ processing of data standardized by Appendix A.

3.PCA

Based on the FPKM method, the dimensionality reduction of data was performed by a PCA unsupervised algorithm, and visualization was carried out to present the repeatability between groups, as shown in Appendix A.

##### Difference Analysis

The R-packet edgeR was used to calculate the expression difference of the expressed count matrix based on the grouping information.

Volcano Map

The difference in mRNA was visualized, as shown in Appendix A.

2.Heatmap

Different mRNA values corresponding to FPKM values were selected for the clustering of heatmap visualization.

3.Gene Function Search

The National Center for Botechnology Information (NCBI) Gene database (www.ncbi.nlm.nih.gov/gene/ (accessed on 26 July 2022)) was used to search the functions of the top 10 differentially expressed genes.

##### GO and Pathway

The differential miRNA target genes were analysed by GO and pathway with the assistance of the R package. GO refers to the international standard classification system of gene function. GO consists of three parts: molecular function (Molecular Function), biological process (Biological Process) and cellular composition (Cellular Component). Through GO enrichment analysis, it is possible to identify not only the important function that leads to the change in traits but also the corresponding genes. The above shows a group of differential GO enrichment analyses. KEGG (Kyoto Encyclopedia of Genes and Genomes) is a bioinformatics resource required to understand biological functions from a genomic perspective. As a multispecies, integrated resource consisting of genomics, chemistry and network information, KEGG cross-references a number of external databases, including a complete set of construction modules (genes and molecules) and wiring diagrams (biomass pathways) to represent cell functions. KEGG involves a set of databases: P A THW A Yjens genes/Sequence Similarity Database (SSDB), biomolecule relationships in information transmission and expression (BRITE), and LIGAND, which is COMPOUND, DRUG, GL YCAN, REACTION, REP AIR and enzyme. According to the enrichment analysis, the method adopted was Fisher accurate inspection, and the data packet was clusterProfiler from R/bioconductor.

##### Gene Set Enrichment Analysis (GSEA)

The differential miRNA target genes were analysed by GO and pathway with the assistance of the R package. Gene Ontology (GO) refers to the international standard classification system of gene function. GO consists of three parts: molecular function (Molecular Function), biological process (Biological Process) and cellular composition (Cellular Component). Through GO enrichment analysis, it is possible to identify the important functions that lead to changes in traits and the corresponding genes. The above shows a group of differential GO enrichment analyses. KEGG (Kyoto Encyclopedia of Genes and Genomes) is a bioinformatics resource required to understand biological functions from a genomic perspective. As a multispecies, integrated resource consisting of genomics, chemistry and network information, it cross-references a number of external databases, including a complete set of construction modules (genes and molecules) and wiring diagrams (biomass pathways) to represent cell functions. KEGG involves a set of databases: P A THW A Yjens genes/Sequence Similarity Database (SSDB), biomolecule relationships in information transmission and expression (BRITE), and LIGAND, which is COMPOUND, DRUG, GL YCAN, REACTION, REP AIR and enzyme. According to the enrichment analysis, the method adopted was Fisher accurate inspection, and the data packet was clusterProfiler from R/Bioconductor, as shown in Appendix A.

### 5.9. Statistical Analysis

Statistical analyses were performed by using GraphPad Prism software (version 9.0). Normality tests were conducted to examine whether all statistical data conformed to a normal distribution. To compare the differences between the two groups, *t*-tests were conducted. When the data showed no conformance to a normal distribution, the Mann-Whitney-Wilcoxon test was carried out to compare the difference between groups, not the *t*-test. One-way analysis of variance (ANOVA) was conducted for statistical comparison between three or more groups. The significant differences between groups are represented by * *p* < 0.05.

## Figures and Tables

**Figure 1 toxins-14-00829-f001:**
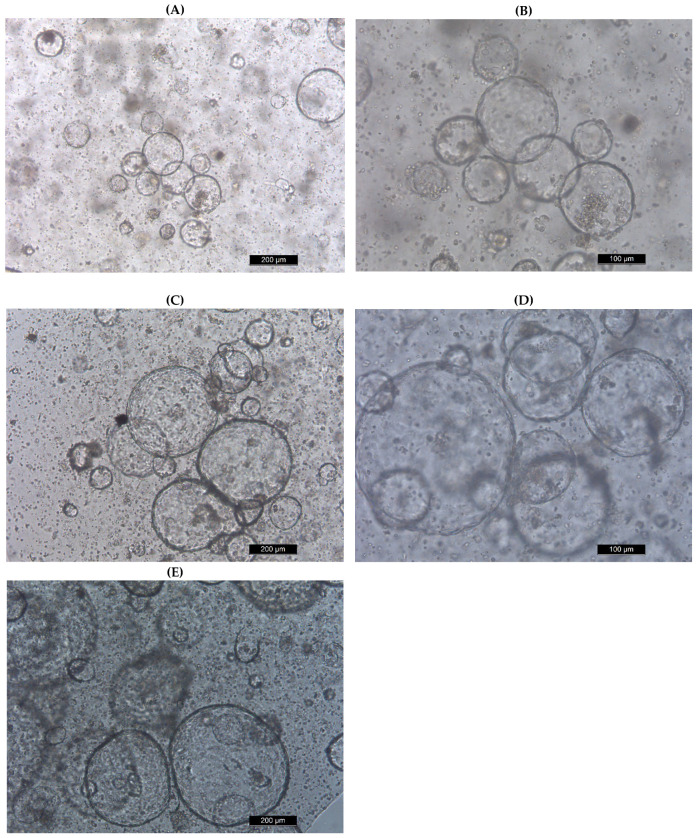
Mouse intestinal organoids culture. (**A**,**B**) Day 1; (**C**,**D**) Day 3; (**E**) Day 5.

**Figure 2 toxins-14-00829-f002:**
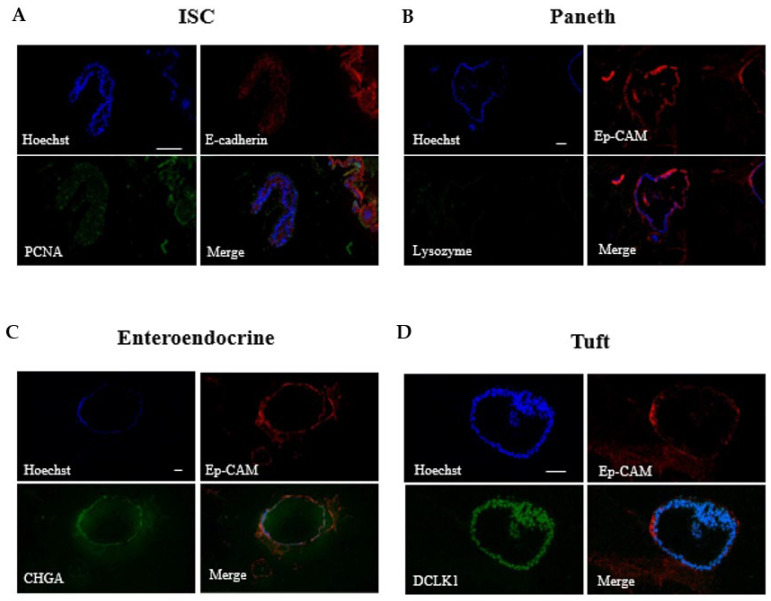
Immunofluorescence of the small intestinal organoids of mice cultured for 10 days. (**A**–**D**) Epithelial cells, positive Ep-CAM and E-cadherin staining, (**A**) stem cells (intestinal stem cells), positive for PCNA staining, (**B**) Paneth cells, positive for lysozyme, (**C**) endocrine cells, positive for Chromogranin A staining, (**D**) clusters of cells (Tuft cells), positive for double cortical protein-like kinase (Doublecortin like kinase, DCLK)-1. Scale base, 50 μM.

**Figure 3 toxins-14-00829-f003:**
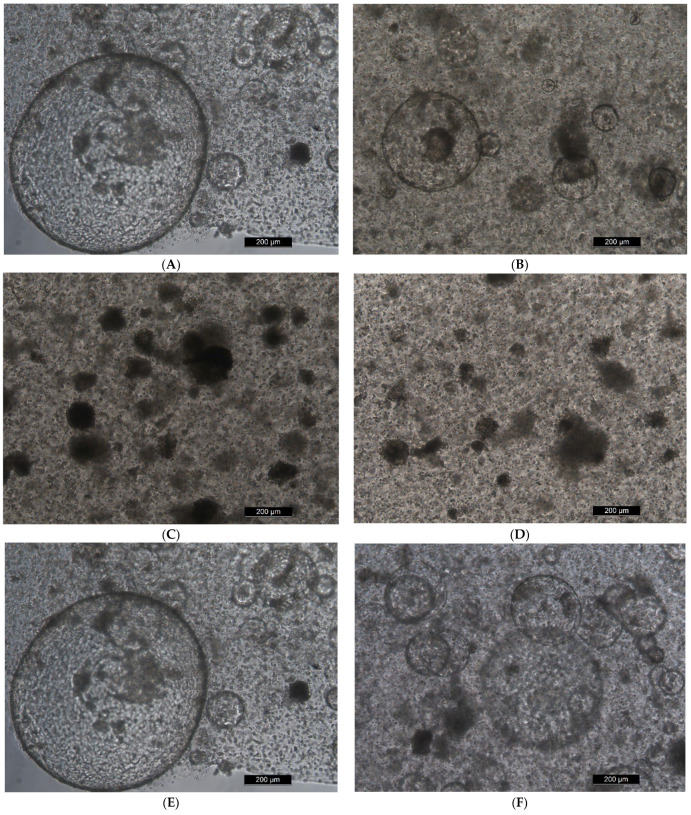
Toxins were added after 14 days of subculture of mouse intestinal organoids. (**A**–**D**) Two days after adding OA, the concentrations were 0, 1, 5 and 10 μM, respectively; (**E**–**H**) two days after adding CgTx, the concentrations were 0, 0.4, 4 and 20 μM, respectively.

**Figure 4 toxins-14-00829-f004:**
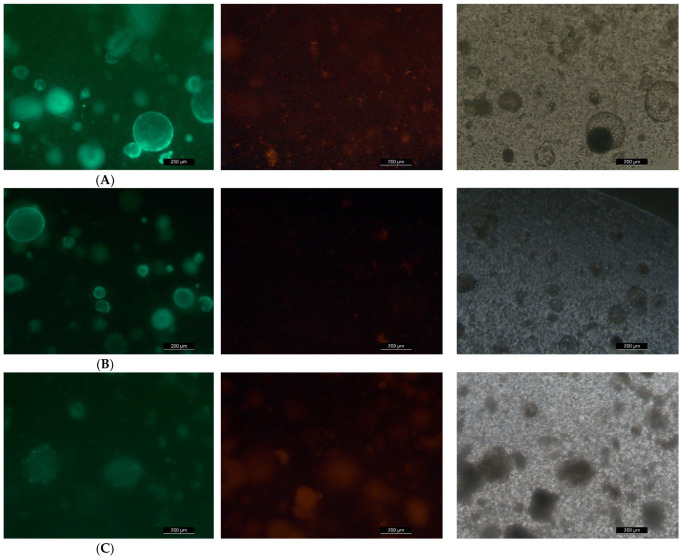
Mortality 2 days after OA was added to the mouse intestinal organoids, (**A**–**D**) AM/PI photos, the concentrations were 0, 1, 5 and 10 μM, respectively; (**E**) mortality, one-way ANOVA, *p* < 0.01, IC_50_ was 3.611 μM; mortality 2 days after CgTx was added to the mouse intestinal organoids, (**F**–**I**) AM/PI photos, the concentrations were 0, 0.4, 4 and 20 μM, respectively; (**J**) mortality, one-way ANOVA, *p* < 0.0001, IC 50 was 7.474 μM; * means *p* < 0.05, **means *p* < 0.01, **** means *p* < 0.0001.

**Figure 5 toxins-14-00829-f005:**
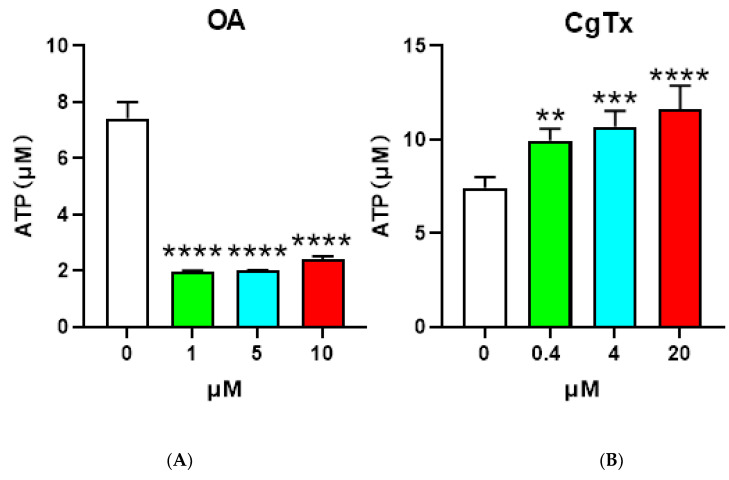
ATPase assay of 2 days after added (**A**) OA; (**B**) CgTx of mouse intestinal organoids via CellTiter-Glo^®^ (Promega), one-way ANOVA, (**A**,**B**) *p* < 0.0001; ** means *p* < 0.01, *** means *p* < 0.001, **** means *p* < 0.0001.

**Figure 6 toxins-14-00829-f006:**
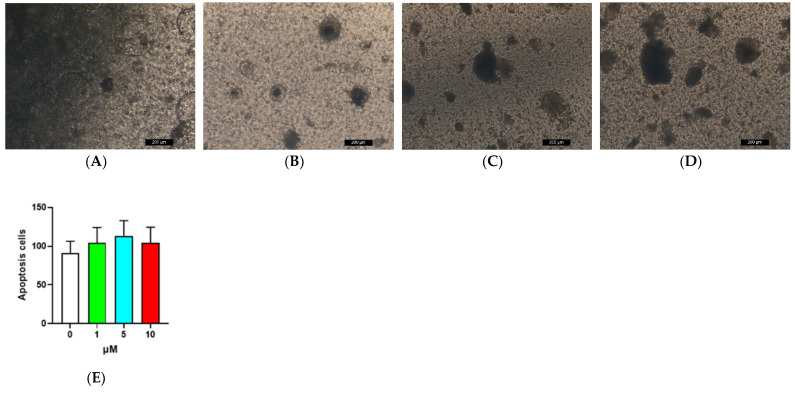
TUNEL assay of 2 days after OA was added to the mouse intestinal organoids, (**A**–**D**) TUNEL photos, the concentrations were 0, 1, 5 and 10 μM, respectively; (**E**) count of apoptosis cells, one-way ANOVA, *p* > 0.05; TUNEL assay of 2 days after CgTx was added to the mouse intestinal organoids, (**F**–**I**) TUNEL photos, the concentrations were 0, 0.4, 4 and 20 μM, respectively; (**J**) count of apoptosis cells, the fluorescence images obtained were not of adequate resolution to include, one-way ANOVA, *p* > 0.05.

**Figure 7 toxins-14-00829-f007:**
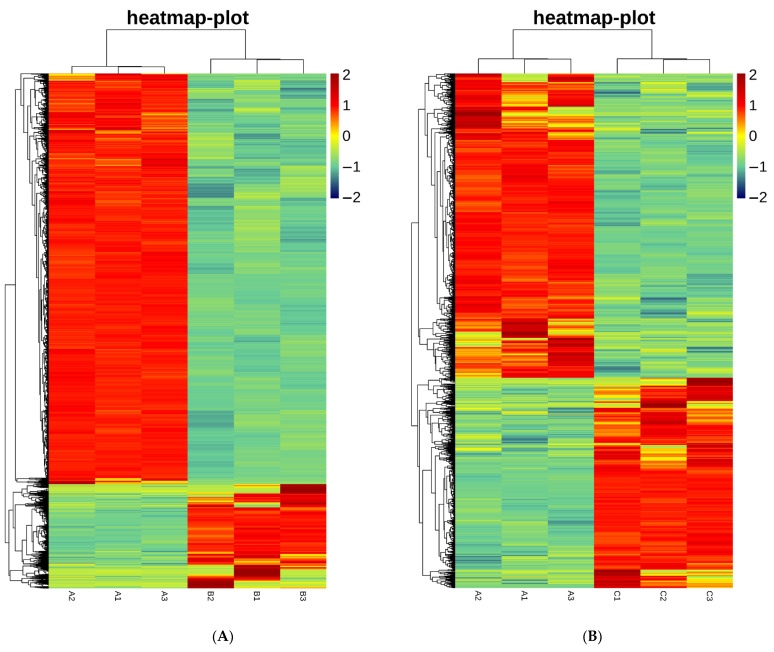
Heatmap of after 14 days of subculture of mouse intestinal organoids, different mRNA values corresponding to the box map of all gene expression values (FPKM) were selected for the clustering of heatmap visualization, (**A**) added with OA for 2 days, and the concentration was 5 μM; (**B**) added with CgTx for 2 days, and the concentration was 4 μM.

**Table 1 toxins-14-00829-t001:** Differential gene expression analysis of RNA-seq of mouse intestinal organoids were subcultured for 14 days and added OA for 2 days, concentration 5 μM.

UpDown	Type	GeneSymbol
DOWN	protein_coding	Ngfr
DOWN	protein_coding	Ifrd1
DOWN	protein_coding	Ccne1
DOWN	protein_coding	Peg3
DOWN	protein_coding	Klf4
DOWN	protein_coding	Slbp
DOWN	protein_coding	Hmox1
DOWN	protein_coding	Trim28
DOWN	protein_coding	Adamts4
DOWN	protein_coding	Tgfbr1
DOWN	protein_coding	Hnrnph1
DOWN	protein_coding	Carhsp1
DOWN	protein_coding	Slc3a2
DOWN	protein_coding	Zfp296
DOWN	protein_coding	Dnajb9
DOWN	protein_coding	Gadd45b
DOWN	protein_coding	Sqstm1
DOWN	protein_coding	Plagl1
DOWN	protein_coding	Tnfaip3
DOWN	protein_coding	Sirt1

**Table 2 toxins-14-00829-t002:** Differential gene expression analysis of RNA-seq of mouse intestinal organoids were subcultured for 14 days and added CgTx for 2 days, concentration 4 μM.

UpDown	Type	GeneSymbol
DOWN	protein_coding	Peg3
DOWN	protein_coding	Zim1
UP	protein_coding	Fosb
UP	protein_coding	Cdc20
UP	protein_coding	Sulf2
DOWN	protein_coding	Id3
DOWN	protein_coding	Slc16a12
DOWN	protein_coding	B4galnt2
DOWN	protein_coding	2310057J18Rik
DOWN	protein_coding	Tmprss6
DOWN	protein_coding	Gabrp
DOWN	protein_coding	Cpm
DOWN	protein_coding	Id2
UP	protein_coding	Olfm4
UP	protein_coding	Tgm1
UP	protein_coding	Nr4a1
DOWN	protein_coding	Sult1c2
DOWN	protein_coding	Cldn6
DOWN	protein_coding	Cacna1h
UP	protein_coding	Il33

**Table 3 toxins-14-00829-t003:** GO of mouse intestinal organoids were subcultured for 14 days and added OA for 2 days, concentration 5 μM.

Ontology	Description
CC	mitochondrial inner membrane
CC	organelle inner membrane
CC	mitochondrial protein-containing complex
BP	mitochondrion organization
CC	Golgi apparatus subcompartment
CC	mitochondrial matrix
BP	organophosphate biosynthetic process
BP	nucleoside phosphate metabolic process
BP	nucleotide metabolic process
BP	generation of precursor metabolites and energy

**Table 4 toxins-14-00829-t004:** GO of mouse intestinal organoids were subcultured for 14 days and added CgTx for 2 days, concentration 4 μM.

Ontology	Description
MF	receptor ligand activity
CC	anchored component of membrane
BP	organic acid biosynthetic process
BP	wound healing
CC	collagen-containing extracellular matrix
BP	carboxylic acid biosynthetic process
BP	cell-substrate adhesion
BP	leukocyte migration
BP	cell chemotaxis
BP	negative regulation of locomotion

**Table 5 toxins-14-00829-t005:** KEGG analysis of mouse intestinal organoids were subcultured for 14 days and added OA for 2 days, concentration 5 μM.

Description
Chemical carcinogenesis–reactive oxygen species
Prion disease
Peroxisome
Pathways of neurodegeneration–multiple diseases
Thermogenesis
Alzheimer disease
Salmonella infection
Parkinson disease
Oxidative phosphorylation
TNF signaling pathway

**Table 6 toxins-14-00829-t006:** KEGG analysis of mouse intestinal organoids were subcultured for 14 days and added CgTx for 2 days, concentration 4 μM.

Description
Cytokine-cytokine receptor interaction
Calcium signaling pathway
Cholesterol metabolism
Arachidonic acid metabolism
Mineral absorption
Hippo signaling pathway
Axon guidance
PPAR signaling pathway
Gastric acid secretion
Sulfur metabolism

## Data Availability

All authors named have participated in the work in a substantive way and are prepared to take public responsibility for the work. The manuscript being submitted to this journal has never been published and that it is not being submitted for publication elsewhere.

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
