# Peer review of "Effects of Various Marine Toxins on the Mouse Intestine Organoid Model"

_toxins, 2022, doi:10.3390/toxins14120829_

Round 1

Reviewer 1 Report

The paper entitled “Effects of Various Marine Toxin on Mouse Intestine Organoids Model” describes the construction of an MIO model to investigate the effects of different concentrations of okadaic acid and conotoxin on cell mortality, ATPase activity, and apoptosis in MIO. And based on these experimental facts, authors used high-throughput RNA-Seq to detect differential genes, GO and pathway, and GSEA in MIO that were gaga cloned to okadaic acid and conotoxin. It should be noted that studying the effects of deep-sea toxins okadaic acid and conotoxin on the digestive system through the mIO model is a promising and effective solution. Based on the results of this study, mIO can be applied as an effective bio-experimental model for deep-sea toxicants okadaic acid and conotoxin.   I think the paper includes some interesting results, but the following points should be cleared before acceptance of the manuscript.

(1) The authors used CTX as an abbreviation of conotoxin in the manuscript. However, in the field of marine toxins, many readers think that CTX means “ciguatoxin”. This isn't very clear.   So, I recommend the authors reconsider the abbreviation.   For example, some authors use CgTx as an abbreviation of conotoxin (Biochemical and Biophysical Research Communications, 409 (2011) 645-650.

(2)  It should be stated why okadaic acid and conotoxin were chosen as marine toxins in this experiment.

(3)  If the authors wanted to know the toxicity of marine toxins in the gut, one wonders why they did not conduct experiments with human organoids, not mouse ones

(4)  The readers in the field of toxins are unfamiliar with organoids, so the authors should explain in detail in the paper results of the construction of MIO model and identification of MIO.

(5)  Some experimental sections are written concisely, and some content is not understandable. Details should be provided.

Author Response

Dear Editors and Reviewers:

Thank you for your letter and for the reviewers’ comments concerning our manuscript entitled “Effects of Various Marine Toxin on Mouse Intestine Organoids Model” (ID: toxins-1923295).Those comments are all valuable and very helpful for revising and improving our paper, as well as the important guiding significance to our researches. We have studied comments carefully and have made correction which we hope meet with approval. Revised portion are marked in red in the paper. The main corrections in the paper and the responds to the reviewer’s comments are as flowing:

Responds to the reviewer’s comments:

Reviewer #1:

  1. Response to comment: (1) The authors used CTX as an abbreviation of conotoxin in the manuscript. However, in the field of marine toxins, many readers think that CTX means “ciguatoxin”. This isn't very clear.   So, I recommend the authors reconsider the abbreviation.   For example, some authors use CgTx as an abbreviation of conotoxin (Biochemical and Biophysical Research Communications, 409 (2011) 645-650.

Response: We are very sorry for our negligence of using CTX as an abbreviation of conotoxin, and we have revised the abbreviation of conotoxin using CgTx, marked in yellow in revised paper.

  1. Response to comment: (2) It should be stated why okadaic acid and conotoxin were chosen as marine toxins in this experiment.

Response: We have made correction according to the Reviewer’s comments, and revised as follow:

Line 70 “In this study, the MIO model was successfully constructed to explore the effects of different concentrations of OA and CgTx (μ-conotoxin Cnâ…¢C), as the representative toxins of DSP and biological polypeptide toxin from marine toxins, on cell mortality, ATPase activity and apoptosis in MIO. Then, based on the above experimental results, high-throughput RNA-Seq was used to detect the differentially expressed genes, GO terms and pathways and Gene Set Enrichment Analysis (GSEA) in MIOs exposed to OA and CgTx. Notably, it is a promising and effective solution to study the effects of the marine toxins OA and CgTx on the digestive system through the MIO model. According to our research results, MIO is applicable as an effective biological experimental model for marine toxins OA and CgTx.” Was added.

  1. Response to comment: (3) If the authors wanted to know the toxicity of marine toxins in the gut, one wonders why they did not conduct experiments with human organoids, not mouse ones

Response: As Reviewer suggested that why we did not conduct experiments with human organoids. Because the acquisition of human tissue had to spend a lot of time to go through all the ethical and legal reviews.

  1. Response to comment: (4) The readers in the field of toxins are unfamiliar with organoids, so the authors should explain in detail in the paper results of the construction of MIO model and identification of MIO.

Response: We have re-written this part according to the Reviewer’s suggestion. Line 81 “2.1. Construction of MIO model

The MIO model shown in Figure 1 was constructed using the medium of SHANGHAI OUTDO BIOTECH and Matrigel and then identified by immunofluorescence experiments to illustrate the epithelial cells, intestinal stem cells, Paneth cells, endocrine cells and tuft cells (Figure 2). The results suggest the successful construction of the MIO model. “ and other detail methods of construction of MIO model and identification of MIO were displayed in Materials and Methods.

  1. Response to comment: (5) Some experimental sections are written concisely, and some content is not understandable. Details should be provided.

Response: Considering the Reviewer’s suggestion, we have re-written this part and here we did not list the changes one by one but marked in yellow in revised paper.

Special thanks to you for your good comments.

We tried our best to improve the manuscript and made some changes in the manuscript. These changes will not influence the content and framework of the paper. And here we did not list the changes but marked in yellow in revised paper.

We appreciate for Editors/Reviewers’ warm work earnestly, and hope that the correction will meet with approval.

Once again, thank you very much for your comments and suggestions.

Reviewer 2 Report

The article describes some effects of okadaic acid and conotoxin on mouse intestine organoids model that may be useful in understanding their toxicity in the digestive system. However, the work needs to be completely rewritten to be suitable for publication.

Author Response

Dear Editors and Reviewers:

Thank you for your letter and for the reviewers’ comments concerning our manuscript entitled “Effects of Various Marine Toxin on Mouse Intestine Organoids Model” (ID: toxins-1923295).Those comments are all valuable and very helpful for revising and improving our paper, as well as the important guiding significance to our researches. We have studied comments carefully and have made correction which we hope meet with approval. Revised portion are marked in red in the paper. The main corrections in the paper and the responds to the reviewer’s comments are as flowing:

Responds to the reviewer’s comments:

Reviewer #2:

Response to comment: The article describes some effects of okadaic acid and conotoxin on mouse intestine organoids model that may be useful in understanding their toxicity in the digestive system. However, the work needs to be completely rewritten to be suitable for publication.

Response: Considering the Reviewer’s suggestion, we have re-written this part completely, and here we did not list the changes one by one but marked in yellow in revised paper. Special thanks to you for your good comments.

We tried our best to improve the manuscript and made some changes in the manuscript. These changes will not influence the content and framework of the paper. And here we did not list the changes but marked in red in revised paper.

We appreciate for Editors/Reviewers’ warm work earnestly, and hope that the correction will meet with approval.

Once again, thank you very much for your comments and suggestions.

Reviewer 3 Report

In the present manuscript, the authors report on a study carried out with marine toxins using a mouse intestine organoids (MIO) as model. The study reveals interesting details, but a revision is necessary. 

- So far, many conotoxins have been described, the authors must specify with which of them the study has been carried out.

- A section of “Conclusions” should be included. This section should describe more clearly the remarkable achievements of this study and the significance of the results.

 - It should be better justified because two toxins with different mechanisms of action and also different chemical structures are used for this study, with the same model.

- The acronym CTX is widely used for marine toxins with a ciguatoxin structure. Modify the acronym throughout the manuscript. Indicate specifically which conotoxin was used and apply the appropriate acronym for the conotoxin used in the trials. In addition, the origin of the okadaic acid used must be indicated, as well as that of this conotoxin.

 - The term “marine toxins” should be used instead of “sea-deep toxin”.

 - Tables 7-10 are minor relevance to the main text of the manuscript and should be included in the supplementary material. In addition, their quality should be improved since they are difficult to read.

 - Figure 7 should describe what A1, A2, etc... refers to.

 - Figure 8 is illegible. Improve its quality and consider its inclusion in a fractionate way in the supplementary material.

 - Results described in section 4.8.4 should be included/weighted/discussed also in section "2 Results".

  - Figure S1 is not cited in the text.

 - The quality of figures S5-S15 should be improved. The information that each one should also be adequately described in the figure captions.

Author Response

Dear Editors and Reviewers:

Thank you for your letter and for the reviewers’ comments concerning our manuscript entitled “Effects of Various Marine Toxin on Mouse Intestine Organoids Model” (ID: toxins-1923295).Those comments are all valuable and very helpful for revising and improving our paper, as well as the important guiding significance to our researches. We have studied comments carefully and have made correction which we hope meet with approval. Revised portion are marked in red in the paper. The main corrections in the paper and the responds to the reviewer’s comments are as flowing:

Responds to the reviewer’s comments:

Reviewer #3:

  1. Response to comment: - So far, many conotoxins have been described, the authors must specify with which of them the study has been carried out.

Response: We are very sorry for our negligence of Indicating specifically which conotoxin was used in the experiments. And we have be indicated in line 295 “5.4. Preparation and exposure of marine toxins

The concentration gradient of the MIO exposed to two kinds of toxins was designed as follows. The low, middle and high concentrations of OA (Puhuashi Technology Development Co., Ltd., A-OA-C100) were 0.1, 5 and 10 μM, respectively. The low, middle and high concentrations of CgTx (μ-conotoxin Cnâ…¢C) (APeptide Co., Ltd., P201028-K6, Sunnyvale, CA, USA) were 0.4, 4 and 20 μM, respectively. MIO was inoculated into 96-well plates and then exposed according to the above concentration gradient after 14 days of subculture. The exposure time was set to 48 hours, and the control group was set up at the same time [1, 7, 10, 12, 13, 17, 34, 42-45].

  1. Response to comment: - A section of “Conclusions” should be included. This section should describe more clearly the remarkable achievements of this study and the significance of the results.

  1. Response: Considering the Reviewer’s suggestion, we have revised in line 203,”“ Conclusions

In summary, an MIO model was constructed to explore the effects of the marine toxins OA and CgTx on MIO. According to the research results, the cell mortality caused by the two toxins at middle and high concentrations was significantly higher compared to the control group, while the activity of ATPase in each group exposed to OA was significantly lower than the activity of ATPase in the control group, while all the CgTx groups were significantly higher than the ATPase activity of the control group, while the number of apoptotic cells was not significantly increased compared with the control group. Through RNA-Seq differential genes, GO and pathway analysis, and GSEA experimental results, OA was discovered to reduce cell metabolism and energy production by affecting cell transcription in MIO, which led to cell death, and that CgTx upregulated the intracellular hormone metabolism pathway by affecting the nuclear receptor pathway of MIO, which led to cell death and the generation of energy in large amounts.

  1. Response to comment: - It should be better justified because two toxins with different mechanisms of action and also different chemical structures are used for this study, with the same model.

Response: Considering the Reviewer’s suggestion, we have revised as follow:

Line 70 “In this study, the MIO model was successfully constructed to explore the effects of different concentrations of OA and CgTx (μ-conotoxin Cnâ…¢C), as the representative toxins of DSP and biological polypeptide toxin from marine toxins, on cell mortality, ATPase activity and apoptosis in MIO. Then, based on the above experimental results, high-throughput RNA-Seq was used to detect the differentially expressed genes, GO terms and pathways and Gene Set Enrichment Analysis (GSEA) in MIOs exposed to OA and CgTx. Notably, it is a promising and effective solution to study the effects of the marine toxins OA and CgTx on the digestive system through the MIO model. According to our research results, MIO is applicable as an effective biological experimental model for marine toxins OA and CgTx.” Was added.

  1. Response to comment: - The acronym CTX is widely used for marine toxins with a ciguatoxin structure. Modify the acronym throughout the manuscript. Indicate specifically which conotoxin was used and apply the appropriate acronym for the conotoxin used in the trials. In addition, the origin of the okadaic acid used must be indicated, as well as that of this conotoxin.

Response: We are very sorry for our negligence of using CTX as an abbreviation of conotoxin, and we have revised the abbreviation of conotoxin using CgTx, marked in yellow in revised paper. In addition, the origin of the okadaic acid and conotoxin have be indicated in line 295 “5.4. Preparation and exposure of marine toxins

The concentration gradient of the MIO exposed to two kinds of toxins was designed as follows. The low, middle and high concentrations of OA (Puhuashi Technology Development Co., Ltd., A-OA-C100) were 0.1, 5 and 10 μM, respectively. The low, middle and high concentrations of CgTx (μ-conotoxin Cnâ…¢C) (APeptide Co., Ltd., P201028-K6, Sunnyvale, CA, USA) were 0.4, 4 and 20 μM, respectively. MIO was inoculated into 96-well plates and then exposed according to the above concentration gradient after 14 days of subculture. The exposure time was set to 48 hours, and the control group was set up at the same time [1, 7, 10, 12, 13, 17, 34, 42-45].“

  1. Response to comment: - The term “marine toxins” should be used instead of “sea-deep toxin”.

Response: We are very sorry for our incorrect writing “sea-deep toxin”, and we have revised the writing “marine toxins”, and marked in yellow in revised paper.

  1. Response to comment:  - Tables 7-10 are minor relevance to the main text of the manuscript and should be included in the supplementary material. In addition, their quality should be improved since they are difficult to read.

Response: As Reviewer suggested that Tables 7-10 are minor relevance to the main text of the manuscript and should be included in the supplementary material. So, we have removed them to supplementary material and improved.

  1. Response to comment:  - Figure 7 should describe what A1, A2, etc... refers to.

Response: Considering the Reviewer’s suggestion, we have revised the describe of Figure 7 in Figure file.

  1. Response to comment: - Figure 8 is illegible. Improve its quality and consider its inclusion in a fractionate way in the supplementary material.

Response: Considering the Reviewer’s suggestion, we have revised the describe of Figure 8 in m Figure file.

  1. Response to comment: - Results described in section 4.8.4 should be included/weighted/discussed also in section "2 Results".

Response: Considering the Reviewer’s suggestion, we have revised in line 113, “2.5.1. Differential gene analysis

The original result of Illumina sequencing is the original image file. Then, after base recognition and error filtering, the original sequencing fragment used for analysis is obtained, referred to as reads. Data Q-value box chart (box plot): A box chart is a statistical chart intended to show the distribution of data. The Q-value box chart is the quality quantile chart. The lowest edge of the yellow rectangle is the Q-value 1/4 quantile, while the upper and lower black lines account for 3/4 of the corresponding quality values, respectively. The blue lines represent the average value of the mass. Different background colours indicate the quality of this part, the green background represents the high quality value part, the orange background indicates the reasonable quality value part, and the red background represents the low quality value part, as shown in Figure S5. Base distribution map: According to the principle of base complementary pairing, GC and AT base pairs conform to a uniform distribution, and the base distribution balance is evaluated against the base distribution map. Due to the random primers used to construct the library, the first few bases will show a preference in nucleotide composition to some extent, thus resulting in normal fluctuations. Then, it tends to stabilize, as shown in Figure S6. The original result of Illumina sequencing is the original image file. Then, after base recognition and error filtering, the original sequencing fragment used for analysis is obtained, which is referred to as reads. Stored in fastq format, the result includes the base composition information of the sequence and its corresponding information about sequence quality. Double-ended sequencing (PE) was divided into two read files: _ R1MagneR2 (Figure S7). The preprocessed sequencing sequence was used for genomic mapping with the assistance of HISAT2 software. Based on global and local search methods, HISAT2 can be applied to carry out mapping efficiently and perform comparisons with the spliced reads in RNA-Seq sequencing data effectively with the default parameter set. The R-packet edgeR was employed to calculate the expression difference of the expression count matrix according to the grouping information (Table 1-2.). In addition, the different mRNAs corresponding to fragments per kilobase of transcript per million mapped reads (FPKM) values were selected for clustering heatmap visualization (Figure 7). The results are listed in Table 1-2.“

  1. Response to comment: -Figure S1 is not cited in the text.

Response: We are very sorry for our negligence of Figure S1, and we have cited it in line 87 “2.2. Detection of MIO activity after exposure to marine toxins

The MIO model was adopted to detect the effect of marine toxins on the activity of MIO (Figure 3). The data on cell mortality were gathered after exposure to marine toxins in the MIO. After the introduction of OA for 2 days, the cell mortality of the 5 μM and 10 μM groups was found to be significantly different from the mortality of the control group (p < 0.01, p < 0.01). For the one-way analysis of variance (ANOVA), p < 0.01, and IC50 was 3.611 μM. After the introduction of CgTx for 2 days, there was a significant difference found in cell mortality between the 4-μM and 20-μM groups and the control group (p < 0.05, p < 0.0001). One-way ANOVA, p < 0.0001, IC50 was 7.474 μM (Figure 4, S1-S2).”

  1. Response to comment: - The quality of figures S5-S15 should be improved. The information that each one should also be adequately described in the figure captions.

Response: Considering the Reviewer’s suggestion, we have revised the describe of Figure S5-S16 in m Supplementary Materials file.

Special thanks to you for your good comments.

We tried our best to improve the manuscript and made some changes in the manuscript. These changes will not influence the content and framework of the paper. And here we did not list the changes but marked in red in revised paper.

We appreciate for Editors/Reviewers’ warm work earnestly, and hope that the correction will meet with approval.

Once again, thank you very much for your comments and suggestions.

Round 2

Reviewer 1 Report

The revisions made by the authors are satisfactory.   However, the qualities of some figures are not good enough.   For example, Fig. 8, S5, S6, S7, S8, S9, S10, and S11 should be improved.  I could not read some text in the figures.   These should be improved before publication.

Author Response

Dear Editors and Reviewers:

Thank you for your letter and for the reviewers’ comments concerning our manuscript entitled “Effects of Various Marine Toxin on Mouse Intestine Organoids Model” (ID: toxins-1923295).Those comments are all valuable and very helpful for revising and improving our paper, as well as the important guiding significance to our researches. We have studied comments carefully and have made correction which we hope meet with approval. Revised portion are marked in red in the paper. The main corrections in the paper and the responds to the reviewer’s comments are as flowing:

Responds to the reviewer’s comments:

Reviewer #1:

  1. Response to comment: (1) The revisions made by the authors are satisfactory.   However, the qualities of some figures are not good enough.   For example, Fig. 8, S5, S6, S7, S8, S9, S10, and S11 should be improved.  I could not read some text in the figures.   These should be improved before publication.

Response: We have made correction according to the Reviewer’s comments, and improved the quality of Fig. 8, S5, S6, S7, S8, S9, S10, and S11.

Special thanks to you for your good comments.

We tried our best to improve the manuscript and made some changes in the manuscript. These changes will not influence the content and framework of the paper. And here we did not list the changes but marked in yellow in revised paper.

We appreciate for Editors/Reviewers’ warm work earnestly, and hope that the correction will meet with approval.

Once again, thank you very much for your comments and suggestions.

Reviewer 3 Report

Dear Editor,

After a reading of the revised manuscript, some details should be modified:

-      As in the rest of the figures, in Figure 2 a letter must be included in the images in order to be properly identified.

-      In Figure 5B, CTX should be changed to CgTx

-      Figure 8 remains unreadable and has poor quality.

-      Authors should consider dividing it into several images in order to increase the font size and make it understandable.

-      In Figure S2 caption CTX should be changed to CgTx.

-      Figures S7, S8, S9, S10, S11, S12, S13 have poor quality. They are unreadable. The font size should be increased, and an appropriate figure caption should be included that describes the image and identifies all elements in the figure.

-      Figure S4. The font size should be increased, and an appropriate figure caption should be included that describes the image and identifies all elements in the figure, and CTX should be changed to CgTx.

Author Response

Dear Editors and Reviewers:

Thank you for your letter and for the reviewers’ comments concerning our manuscript entitled “Effects of Various Marine Toxin on Mouse Intestine Organoids Model” (ID: toxins-1923295).Those comments are all valuable and very helpful for revising and improving our paper, as well as the important guiding significance to our researches. We have studied comments carefully and have made correction which we hope meet with approval. Revised portion are marked in red in the paper. The main corrections in the paper and the responds to the reviewer’s comments are as flowing:

Responds to the reviewer’s comments:

Reviewer #3:

  1. Response to comment: -As in the rest of the figures, in Figure 2 a letter must be included in the images in order to be properly identified.

Response: We are very sorry for our negligence of letter not in images if Figure 2. And we have be added them.

  1. Response to comment: -In Figure 5B, CTX should be changed to CgTx.

Response: Considering the Reviewer’s suggestion, we have changed CTX to CgTx in Figure 5B.

  1. Response to comment: -Figure 8 remains unreadable and has poor quality.

Response: Considering the Reviewer’s suggestion, we have improved quality of Figure 8.

  1. Response to comment: -Authors should consider dividing it into several images in order to increase the font size and make it understandable.

Response: We are very sorry for our negligence of images quality, in addition, we have improved quality of the whole Figures.

  1. Response to comment: -In Figure S2 caption CTX should be changed to CgTx.

Response: Considering the Reviewer’s suggestion, we have changed CTX to CgTx in Figure S2.

  1. Response to comment:  -Figures S7, S8, S9, S10, S11, S12, S13 have poor quality. They are unreadable. The font size should be increased, and an appropriate figure caption should be included that describes the image and identifies all elements in the figure.

Response: We are very sorry for our negligence of images quality, in addition, we have improved quality of the whole Figures above.

  1. Response to comment:  -      Figure S4. The font size should be increased, and an appropriate figure caption should be included that describes the image and identifies all elements in the figure, and CTX should be changed to CgTx.

Response: Considering the Reviewer’s suggestion, we have revised the describe of Figure S4, and changed CTX to CgTx.

Special thanks to you for your good comments.

We tried our best to improve the manuscript and made some changes in the manuscript. These changes will not influence the content and framework of the paper. And here we did not list the changes but marked in red in revised paper.

We appreciate for Editors/Reviewers’ warm work earnestly, and hope that the correction will meet with approval.

Once again, thank you very much for your comments and suggestions.
